# NeuroEvolve: A Dynamic Brain Graph Deep Generative Model

**Alexander Campbell** [* 1 2]   **Simeon Spasov** [* 1 3]   **Nicola Toschi** [4 5]   **Pietro Liò** [1]

## Abstract

Graphs are a natural representation of brain activity derived from functional magnetic imaging (fMRI) data. It is well known that communities of nodes extracted from brain graphs, referred to as functional connectivity networks (FCNs), serve as useful biomarkers for understanding brain function and dysfunction. Previous works, however, ignore the temporal dynamics of the brain and focus on static graph representations. In this paper we propose NeuroEvolve, a dynamic brain graph deep generative model which simultaneously learns graph-, node-, and community-level embeddings in an unsupervised fashion. Specifically, NeuroEvolve represents brain graph nodes as embeddings sampled from a distribution over communities that evolve over time. The community distribution is parameterized using neural networks that learn from subject and node embeddings as well as past community assignments. Experiments on real-world fMRI data demonstrate NeuroEvolve outperforms state-of-the-art baselines in graph generation, dynamic link prediction, and is comparable for graph classification. Finally, an interpretability analysis of the learnt community distributions reveals overlap with known FCNs reported in neuroscience literature.

## 1. Introduction

Functional magnetic resonance imaging (fMRI) is a non-invasive imaging technique primarily used for measur-

ing blood-oxygen level dependent (BOLD) signal in the brain (Huettel et al., 2004). A natural representation of fMRI data is as a discrete-time graph, henceforth referred to as a dynamic brain graph (DBG), consisting of a set of fixed nodes corresponding to anatomically separated brain regions (Lawrence et al., 2021; Hess et al., 2018) and a set of time-varying edges determined by a measure of dynamic functional connectivity (FC) (Calhoun et al., 2014; Hutchison et al., 2013). DBGs have been widely used in graph-based network analysis for understanding brain function (Hirsch & Wohlschlaeger, 2022; Raz et al., 2016) and dysfunction (Alonso Martínez et al., 2020; Dautricourt et al., 2022; Yu et al., 2015).

Recently, there is growing interest in using deep learning-based methods for learning representations of graph-structured data (Goyal & Ferrara, 2018; Hamilton, 2020). A graph representation typically consists of a low-dimensional vector embedding of either the entire graph (Narayanan et al., 2017) or a part of it's structure such as nodes (Grover & Leskovec, 2016), edges (Gao et al., 2019), or subgraphs (Adhikari et al., 2017). Although originally formulated for static graphs, several existing methods have been extended (Mahdavi et al., 2018; Goyal et al., 2020), and new ones proposed (Zhou et al., 2018; Sankar et al., 2020), for dynamic graphs. The embeddings are usually learnt in either a supervised or unsupervised fashion and are typically used in tasks such as node classification (Pareja et al., 2020) and dynamic link prediction (Goyal et al., 2018).

To date, very few deep learning-based methods have been designed for, or existing methods applied to, representation learning of DBGs. Those that do tend to use graph neural networks (GNNs) (Wu et al., 2020) that are designed for learning node- and graph-level embeddings (Kim et al., 2021; Dahan et al., 2021). Although these embeddings are effective at representing local/global graph structure, they are less adept at representing topological structures in-between these two extremes such a clusters of nodes or communities (Wang et al., 2017). Recent methods that explicitly incorporate community embeddings alongside node embeddings have shown improved performance for static graph representation learning tasks (Sun et al., 2019; Cavallari et al., 2017). How to leverage the relatedness of graph, node, and community embeddings in a unified framework for DBG representation learning remains under-explored.

---

[*]Equal contribution [1]Department of Computer Science and Technology, University of Cambridge, Cambridge, United Kingdom [2]The Alan Turing Institute, London, United Kingdom [3]German Center for Neurodegenerative Diseases, Bonn, Germany [4]University of Rome Tor Vergata, Rome, Italy [5]A.A. Martinos Center for Biomedical Imaging, Harvard Medical School, Boston, United States. Correspondence to: Alexander Campbell <ajrc4@cl.cam.ac.uk>, Simeon Spasov <ses88@cl.cam.ac.uk>.

*Workshop on Interpretable ML in Healthcare at International Conference on Machine Learning (ICML)*, Honolulu, Hawaii, USA. 2023. Copyright 2023 by the author(s).

**Contributions** To address these shortcomings, we propose NeuroEvolve[1], a hierarchical structured deep generative model (DGM) specifically designed for unsupervised learning of DBGs derived from multi-subject fMRI data. NeuroEvolve combines graph, node, and community embeddings in a unified framework, utilizing neural networks (NNs) to parameterize a community distribution over the nodes that evolves over time. NeuroEvolve also incorporates inductive biases in its structure inspired from prior knowledge about brain FCNs. We evaluate NeuroEvolve on multiple real-world fMRI datasets and show that it outperforms state-of-the-art baselines for graph reconstruction, dynamic link prediction, and achieves comparable results for graph classification.

## 2. Related work

**Dynamic graph generative models** Classic generative models for graph-structured data are typically designed for modeling a small set of specific properties (e.g. degree distribution, eigenvalues, modularity) of static graphs (Erdos et al., 1960; Barabási & Albert, 1999; Nowicki & Snijders, 2001). DGMs that exploit the learning capacity of NNs are able to learn more expressive graph distributions (Mehta et al., 2019; Kipf & Welling, 2016b; Sarkar et al., 2020). Recent DGMs for dynamic graphs are majority VAE-based (Kingma & Welling, 2013) and are unable to learn community representations (Hajiramezanali et al., 2019; Gracious et al., 2021; Zhang et al., 2021). The few that do, are designed for static graphs (Sun et al., 2019; Khan et al., 2021; Cavallari et al., 2017).

**Learning representations of dynamic brain graphs** BOLD signals derived from fMRI, whether at the voxel or brain region level, represent non stationary timeseries (Guan et al., 2020). As such, how BOLD signals relate to each other spatially changes over time. Within the context of dynamic FC, it is essential to capture these time varying spatial relationships. Most unsupervised representation learning methods for DBGs tend to focus on clustering DBGs into a finite number of connectivity patterns that recur over time (Allen et al., 2014; Spencer & Goodfellow, 2022). Community detection is another commonly used method but is mainly applied to static brain graphs (Pavlović et al., 2020; Esfahlani et al., 2021). Extensions to DBGs are typically not end-to-end trainable and do not scale to multi-subject datasets (Ting et al., 2020; Martinet et al., 2020a). Recent deep learning-based methods are predominately GNN-based (Kim et al., 2021; Dahan et al., 2021). Unlike NeuroEvolve, these methods are supervised and focus on learning deterministic node- and/or graph-level

---

[1]Code available at https://github.com/simeon-spasov/dynamic-brain-graph-deep-generative-model

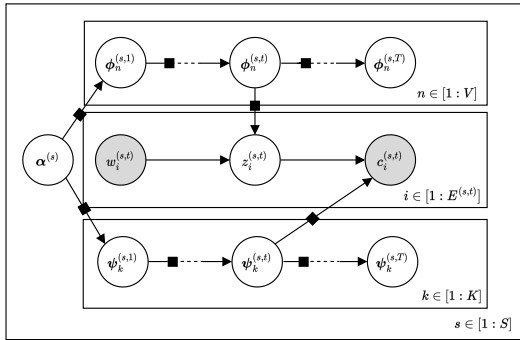

*Figure 1.* Plate diagram summarizing the generative model of NeuroEvolve. Latent and observed variables are denoted by white-and gray-shaded circles, respectively. Solid black squares denote mappings parameterized by a neural network.

representations.

## 3. Problem formulation

We consider a dataset of multi-subject DBGs derived from fMRI data $\mathcal{D} \equiv \mathcal{G}^{(1:S, 1:T)} = \{\mathcal{G}^{(s,t)}\}_{s,\,t=1}^{S,\,T}$ that share a common set of nodes $\mathcal{V} = \{v_1, \ldots, v_V\}$ over $T \in \mathbb{N}$ snapshots for $S \in \mathbb{N}$ subjects. Each $\mathcal{G}^{(s,t)} \in \mathcal{G}^{(1:S, 1:T)}$ denotes a non-attributed, unweighted, and undirected brain graph for the $s$-th subject at the $t$-th snapshot. We define a brain graph snapshot as a tuple $\mathcal{G}^{(s,t)} = (\mathcal{V}, \mathcal{E}^{(s,t)})$ where $\mathcal{E}^{(s,t)} \subseteq \mathcal{V} \times \mathcal{V}$ denotes an edge set. The $i$-th edge for the $s$-th subject at the $t$-th snapshot $e_i^{(s,t)} \in \mathcal{E}^{(s,t)}$ is defined $e_i^{(s,t)} = (w_i^{(s,t)}, c_i^{(s,t)})$ where $w_i^{(s,t)}$ is a source node and $c_i^{(s,t)}$ is a target node.

We assume each node corresponds to a brain region making the number of nodes $|\mathcal{V}| = V \in \mathbb{N}$ fixed over subjects and snapshots. We also assume edges correspond to a measure FC allowing the number of edges $|\mathcal{E}^{(s,t)}| = E^{(s,t)} \in \mathbb{N}$ to vary over subjects as well as snapshots. We further assume there exists $K \in \mathbb{N}$ clusters of nodes, or communities, the membership of which dynamically changes between snapshots for each subject. Let $z_i^{(s,t)} \in [1:K]$ denote the latent community assignment of the $i$-th edge for the $s$-th subject at the $t$-th snapshot.

For each subject's DBG we aim to learn, in an unsupervised fashion, graph $\boldsymbol{\alpha}^{(s)} \in \mathbb{R}^{H_\alpha}$, node $\boldsymbol{\phi}_{1:V}^{(s,t)} = [\boldsymbol{\phi}_n^{(s,t)}] \in \mathbb{R}^{V \times H_\phi}$, and community $\boldsymbol{\psi}_{1:K}^{(s,t)} = [\boldsymbol{\psi}_k^{(s,t)}] \in \mathbb{R}^{K \times H_\psi}$ embeddings of dimensions $H_\alpha, H_\phi, H_\psi \in \mathbb{N}$, respectively, for use in a variety of downstream tasks.

## 4. Method

NeuroEvolve is a hierarchical deep generative model and inference network that accomplishes end-to-end learning of

graph, node, and community embeddings from multi-subject DBG data. NeuroEvolve treats the embeddings and edge community assignments as latent random variables $\Omega^{(s,t)} = \{\boldsymbol{\alpha}^{(s)}, \boldsymbol{\phi}_{1:V}^{(s,t)}, \boldsymbol{\psi}_{1:K}^{(s,t)}, \{z_i^{(s,t)}\}_{i=1}^{E^{(s,t)}}\}$, which along with the observed DBGs, characterize a probabilistic latent variable model with joint density $p_\theta(\mathcal{G}^{1:S,1:T}, \Omega^{1:S,1:T})$.

## 4.1. Generative model

---
**Algorithm 1** NeuroEvolve generative process
---
1: **Input:** Common node set $\mathcal{V}$, source nodes from all edges $\{w_i^{(s,t)} : i = 1, \ldots, E^{(s,t)}\}_{s,t=1}^{S,T}$
2: **Require:** Number of communities $K \in \mathbb{N}$; embedding dimensions $H_\alpha, H_\phi, H_\psi \in \mathbb{N}$; number of layers in NNs $L_\phi, L_\psi, L_z \in \mathbb{N}$; temporal smoothness $\sigma_\psi, \sigma_\phi \in \mathbb{R}_{>0}$
3: $\mathcal{D} \leftarrow \emptyset$
4: **for** $s \leftarrow 1$ to $S$ **do**
5: $\quad \boldsymbol{\alpha}^{(s)} \sim p(\boldsymbol{\alpha}^{(s)}) = \text{Normal}(\mathbf{0}_{H_\alpha}, \mathbf{I}_{H_\alpha})$
6: $\quad$ **for** $t \leftarrow 1$ to $T$ **do**
7: $\qquad$ **for** $k \leftarrow 1$ to $K$ **do**
8: $\qquad\quad$ **if** $t = 1$ **then**
9: $\qquad\qquad \boldsymbol{\psi}_k^{(s,0)} = \text{MLP}_{\theta_\psi}(\boldsymbol{\alpha}^{(s)})$
10: $\qquad\quad$ **end if**
11: $\qquad\quad p(\boldsymbol{\psi}_k^{(s,t)}|\boldsymbol{\psi}_k^{(s,t-1)}) = \text{Normal}(\boldsymbol{\psi}_k^{(s,t-1)}, \sigma_\psi \mathbf{I}_{H_\psi})$
12: $\qquad\quad \boldsymbol{\psi}_k^{(s,t)} \sim p(\boldsymbol{\psi}_k^{(s,t)}|\boldsymbol{\psi}_k^{(s,t-1)})$
13: $\qquad$ **end for**
14: $\qquad$ **for** $n \leftarrow 1$ to $V$ **do**
15: $\qquad\quad$ **if** $t = 1$ **then**
16: $\qquad\qquad \boldsymbol{\phi}_n^{(s,0)} = \text{MLP}_{\theta_\phi}(\boldsymbol{\alpha}^{(s)})$
17: $\qquad\quad$ **end if**
18: $\qquad\quad p(\boldsymbol{\phi}_n^{(s,t)}|\boldsymbol{\phi}_n^{(s,t-1)}) = \text{Normal}(\boldsymbol{\phi}_n^{(s,t-1)}, \sigma_\phi \mathbf{I}_{H_\phi})$
19: $\qquad\quad \boldsymbol{\phi}_n^{(s,t)} \sim p(\boldsymbol{\phi}_n^{(s,t)}|\boldsymbol{\phi}_n^{(s,t-1)})$
20: $\qquad$ **end for**
21: $\qquad \mathcal{E}^{(s,t)} \leftarrow \emptyset$
22: $\qquad$ **for** $i \leftarrow 1$ to $E^{(s,t)}$ **do**
23: $\qquad\quad \text{logit } \hat{\boldsymbol{\pi}}_i^{(s,t)} = \text{MLP}_{\theta_z}(\boldsymbol{\phi}_{w_i}^{(s,t)})$
24: $\qquad\quad p(z_i^{(s,t)}|w_i^{(s,t)}) = \text{Categorical}(\hat{\boldsymbol{\pi}}_i^{(s,t)})$
25: $\qquad\quad z_i^{(s,t)} \sim p(z_i^{(s,t)}|w_i^{(s,t)})$
26: $\qquad\quad \text{logit } \tilde{\boldsymbol{\pi}}_i^{(s,t)} = \text{MLP}_{\theta_c}(\boldsymbol{\psi}_{z_i}^{(s,t)})$
27: $\qquad\quad p_{\theta_c}(c_i^{(s,t)}|z_i^{(s,t)}) = \text{Categorical}(\tilde{\boldsymbol{\pi}}_i^{(s,t)})$
28: $\qquad\quad c_i^{(s,t)} \sim p_{\theta_c}(c_i^{(s,t)}|z_i^{(s,t)})$
29: $\qquad\quad \mathcal{E}^{(s,t)} \leftarrow \mathcal{E}^{(s,t)} \cup \{(w_i^{(s,t)}, c_i^{(s,t)})\}$
30: $\qquad$ **end for**
31: $\qquad \mathcal{G}^{(s,t)} \leftarrow (\mathcal{V}, \mathcal{E}^{(s,t)})$
32: $\qquad \mathcal{D} \leftarrow \mathcal{D} \cup \{\mathcal{G}^{(s,t)}\}$
33: $\quad$ **end for**
34: **end for**
---

**Graph embeddings** The generative process of NeuroEvolve begins by sampling graph embeddings from a prior $\boldsymbol{\alpha}^{(s)} \sim p_{\theta_\alpha}(\boldsymbol{\alpha}^{(s)})$ implemented as a normal distribution

$$p_{\theta_\alpha}(\boldsymbol{\alpha}^{(s)}) = \text{Normal}(\mathbf{0}_{H_\alpha}, \mathbf{I}_{H_\alpha}) \qquad (1)$$

where $\mathbf{0}_{H_\alpha}$ and $\mathbf{I}_{H_\alpha}$ denote a zero matrix a identity matrix, respectively. Each embedding, represented as a $H_\alpha$-dimensional vector $\boldsymbol{\alpha}^{(s)} \in \mathbb{R}^{H_\alpha}$, encapsulates subject-specific information that remains constant over snapshots.

**Node and community embeddings** Next, let $\boldsymbol{\phi}_n^{(s,t)} \in \mathbb{R}^{H_\phi}$ and $\boldsymbol{\psi}_k^{(s,t)} \in \mathbb{R}^{H_\psi}$ denote the $n$-th node and the $k$-th community embedding, respectively. To incorporate temporal dynamics, we assume the node and community embeddings are related through Markov chains with prior transition distributions $\boldsymbol{\phi}_n^{(s,t)} \sim p_{\theta_\phi}(\boldsymbol{\phi}_n^{(s,t)}|\boldsymbol{\phi}_n^{(s,t-1)})$ and $\boldsymbol{\psi}_k^{(s,t)} \sim p_{\theta_\psi}(\boldsymbol{\psi}_k^{(s,t)}|\boldsymbol{\psi}_k^{(s,t-1)})$. We specify each prior to be a normal distribution following

$$p_{\theta_\phi}(\boldsymbol{\phi}_n^{(s,t)}|\boldsymbol{\phi}_n^{(s,t-1)}) = \text{Normal}(\boldsymbol{\phi}_n^{(s,t-1)}, \sigma_\phi \mathbf{I}_{H_\phi}) \quad (2)$$

$$p_{\theta_\psi}(\boldsymbol{\psi}_k^{(s,t)}|\boldsymbol{\psi}_k^{(s,t-1)}) = \text{Normal}(\boldsymbol{\psi}_k^{(s,t-1)}, \sigma_\psi \mathbf{I}_{H_\psi}). \quad (3)$$

The means of each distribution are initialized via NN transformations of the graph embeddings, i.e. $\boldsymbol{\phi}_n^{(s,0)} = \text{MLP}_{\theta_\phi}(\boldsymbol{\alpha}^{(s)})$, $\boldsymbol{\psi}_k^{(s,0)} = \text{MLP}_{\theta_\psi}(\boldsymbol{\alpha}^{(s)})$, where $\text{MLP}_{\theta_j} : \mathbb{R}^{H_\alpha} \to \mathbb{R}^{H_j}$ is a $L_j$-layered multilayer perceptron (MLP) for $j \in \{\phi, \psi\}$. The standard deviations $\sigma_\phi, \sigma_\psi \in \mathbb{R}_{\geq 0}$ are hyperparameters controlling how smoothly each embedding changes between consecutive snapshots.

**Edge generation** We next describe the edge generative process of a graph snapshot $\mathcal{G}^{(s,t)} \in \mathcal{G}^{(1:S,1:T)}$. Similar to Sun et al. (2019), for each edge $e_i^{(s,t)} = (w_i^{(s,t)}, c_i^{(s,t)}) \in \mathcal{E}^{(s,t)}$ we first sample a latent community assignment $z_i^{(s,t)} \in [1 : K]$ from a conditional prior $z_i^{(s,t)} \sim p_{\theta_z}(z_i^{(s,t)}|w_i^{(s,t)})$ implemented as a categorical distribution

$$p_{\theta_z}(z_i^{(s,t)}|w_i^{(s,t)}) = \text{Categorical}(\tilde{\boldsymbol{\pi}}_i^{(s,t)}) \qquad (4)$$

$$\text{logit } \tilde{\boldsymbol{\pi}}_i^{(s,t)} = \text{MLP}_{\theta_z}(\boldsymbol{\phi}_{w_i}^{(s,t)}) \qquad (5)$$

where $\text{MLP}_{\theta_z} : \mathbb{R}^{H_\phi} \to \mathbb{R}^K$ is a $L_z$-layered MLP that parameterizes community probabilities using node embeddings indexed by $w_i^{(s,t)}$. In other words, each source node $w_i^{(s,t)}$ is represented as a mixture of communities.

A linked target node $c_i^{(s,t)} \in [1 : V]$ is then sampled from the conditional likelihood $c_i^{(s,t)} \sim p_{\theta_c}(c_i^{(s,t)}|z_i^{(s,t)})$ which is also implemented as a categorical distribution

$$p_{\theta_c}(c_i^{(s,t)}|z_i^{(s,t)}) = \text{Categorical}(\hat{\boldsymbol{\pi}}_i^{(s,t)}) \qquad (6)$$

$$\text{logit } \hat{\boldsymbol{\pi}}_i^{(s,t)} = \text{MLP}_{\theta_c}(\boldsymbol{\psi}_{z_i}^{(s,t)}) \qquad (7)$$

where $\text{MLP}_{\theta_c} : \mathbb{R}^{H_\psi} \to \mathbb{R}^V$ is a $L_c$-layered MLP that parameterizes node probabilities using community embeddings indexed by $z_i^{(s,t)}$. That is, each community assignment $z_i^{(s,t)}$ is represented as a mixture of nodes. By integrating out the latent community assignment variable

$$p(c_i^{(s,t)}|w_i^{(s,t)}) = \sum_{z_i^{(s,t)} \in [1:K]} p_{\theta_c}(c_i^{(s,t)}|z_i^{(s,t)}) p_{\theta_z}(z_i^{(s,t)}|w_i^{(s,t)}) \quad (8)$$

we define the likelihood of node $c_i^{(s,t)}$ being a linked neighbor of node $w_i^{(s,t)}$, in a given graph snapshot.

**Factorized generative model** Given this generative process, the joint probability of the observed data and the latent variables can be factorized following

$$
\begin{aligned}
p_\theta(\mathcal{G}^{1:S\,1:T},\,\Omega^{1:S,\,1:T}) = \prod_{s=1}^{S} \Bigg( & p_{\theta_\alpha}(\boldsymbol{\alpha}^{(s)}) \\
& \times \prod_{t=1}^{T} \Bigg( \prod_{n=1}^{V} p_{\theta_\phi}(\boldsymbol{\phi}_n^{(s,t)}|\boldsymbol{\phi}_n^{(s,t-1)}) \\
& \times \prod_{k=1}^{K} p_{\theta_\psi}(\boldsymbol{\psi}_k^{(s,t)}|\boldsymbol{\psi}_k^{(s,t-1)}) \quad (9) \\
& \times \prod_{i=1}^{E^{(s,t)}} p_{\theta_z}(z_i^{(s,t)}|\boldsymbol{\phi}_{w_i}^{(s,t)}) \\
& \times p_{\theta_c}(c_i^{(s,t)}|\boldsymbol{\psi}_{z_i}^{(s,t)}) \Bigg) \Bigg)
\end{aligned}
$$

where $\theta = \{\theta_\phi, \theta_\psi, \theta_z, \theta_c\}$ is the set of generative model parameters, i.e. NN weights. See Figure 1 for a graphical representation of NeuroEvolve and Algorithm 1 for a summary of the generative process.

## 4.2. Inference model

Inferring the posterior distribution $p_\theta(\Omega^{(1:S,\,1:T)}|\mathcal{G}^{(1:S,\,1:T)})$ is intractable so we resort to variational inference (Jordan et al., 1999) to approximate the true posterior with a variational distribution $q_\lambda(\Omega^{(1:S,\,1:T)})$. For our training algorithm, we maximize a lower bound on the log marginal likelihood of the DBGs, referred to as the ELBO (**e**vidence **l**ower **bo**und)

$$
\begin{aligned}
\mathcal{L}_{\text{ELBO}}(\theta, \lambda) &= \mathbb{E}_{q_\lambda}\left[ \log \frac{p_\theta(\mathcal{G}^{(1:S,\,1:T)},\,\Omega^{(1:S,\,1:T)})}{q_\lambda(\Omega^{(1:S,\,1:T)})} \right] \\
&\leq \log p_\theta(\mathcal{G}^{(1:S,\,1:T)}) \quad (10)
\end{aligned}
$$

where $\mathbb{E}_{q_\lambda}[\cdot]$ denotes the expectation with respect to the variational distribution $q_\lambda(\Omega^{(1:S,\,1:T)})$. By maximizing the ELBO with respect to the generative and variational parameters $\theta$ and $\lambda$ we train our generative model and perform Bayesian inference, respectively.

**Structured variational distribution** To ensure a good approximation to the true the posterior, we retain the Markov properties of the node and community embeddings resulting in a structured variational distribution (Hoffman & Blei,

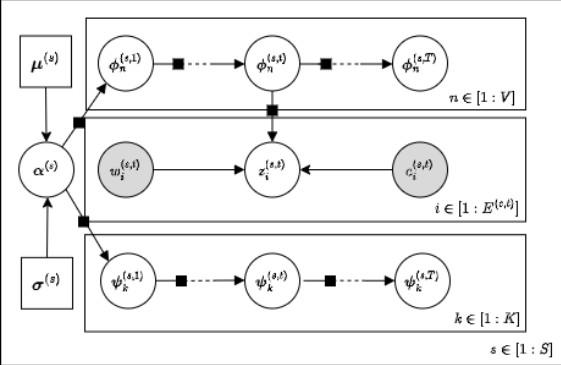

*Figure 2.* Plate diagram summarizing the inference model of NeuroEvolve. Latent and observed variables are denoted by white-and gray-shaded circles, respectively. Solid black squares denote mappings parameterized by a neural network.

2015; Saul & Jordan, 1995) which factorizes following

$$
\begin{aligned}
q_\lambda(\Omega^{(1:S,\,1:T)}) = \prod_{s=1}^{S} \Bigg( & q_{\lambda_\alpha}(\boldsymbol{\alpha}^{(s)}) \\
& \times \prod_{t=1}^{T} \Bigg( \prod_{n=1}^{V} q_{\lambda_\phi}(\boldsymbol{\phi}_n^{(s,t)}|\boldsymbol{\phi}_n^{(s,t-1)}) \\
& \times \prod_{k=1}^{K} q_{\lambda_\psi}(\boldsymbol{\psi}_k^{(s,t)}|\boldsymbol{\psi}_k^{(s,t-1)}) \quad (11) \\
& \times \prod_{i=1}^{E^{(s,t)}} q_{\lambda_z}(z_i^{(s,t)}|\boldsymbol{\phi}_{w_i}^{(s,t)},\,\boldsymbol{\phi}_{c_i}^{(s,t)}) \Bigg) \Bigg).
\end{aligned}
$$

Moreover, we specify each variational distribution to be from the same family of distribution as it's equivalent in the generative model. For the graph embeddings we have

$$
q_{\lambda_\alpha}(\boldsymbol{\alpha}^{(s)}) = \text{Normal}(\boldsymbol{\mu}^{(s)},\,\boldsymbol{\sigma}^{(s)}\mathbf{I}_{H_\alpha}). \quad (12)
$$

Next, for the node embeddings

$$
q_{\lambda_\phi}(\boldsymbol{\phi}_n^{(s,t)}|\boldsymbol{\phi}_n^{(s,t-1)}) = \text{Normal}(\hat{\boldsymbol{\mu}}_n^{(s,t)},\,\hat{\boldsymbol{\sigma}}_n^{(s,t)}\mathbf{I}_{H_\phi}) \quad (13)
$$

$$
\{\hat{\boldsymbol{\mu}}_n^{(s,t)},\,\log\hat{\boldsymbol{\sigma}}_n^{(s,t)}\} = \text{GRU}_{\lambda_\phi}(\boldsymbol{\phi}_n^{(s,t-1)}) \quad (14)
$$

and community embeddings

$$
q_{\lambda_\psi}(\boldsymbol{\psi}_k^{(s,t)}|\boldsymbol{\psi}_k^{(s,t-1)}) = \text{Normal}(\tilde{\boldsymbol{\mu}}_k^{(s,t)},\,\tilde{\boldsymbol{\sigma}}_k^{(s,t)}\mathbf{I}_{H_\psi}) \quad (15)
$$

$$
\{\tilde{\boldsymbol{\mu}}_k^{(s,t)},\,\log\tilde{\boldsymbol{\sigma}}_k^{(s,t)}\} = \text{GRU}_{\lambda_\psi}(\boldsymbol{\psi}_k^{(s,t-1)}) \quad (16)
$$

where $\text{GRU}_{\lambda_j} : \mathbb{R}^{H_j} \to \mathbb{R}^{H_j}$ is a $L_j$-layered gated recurrent unit (GRU) (Cho et al., 2014) for $j \in \{\phi, \psi\}$. Furthermore, we use MLPs to initialize the GRUs with graph embeddings such that $\boldsymbol{\phi}_n^{(s,0)} = \text{MLP}_{\lambda_\phi}(\boldsymbol{\alpha}^{(s)})$ and $\boldsymbol{\psi}_k^{(s,0)} = \text{MLP}_{\lambda_\psi}(\boldsymbol{\alpha}^{(s)})$ where $\text{MLP}_{\lambda_j} : \mathbb{R}^{N_\alpha} \to \mathbb{R}^{N_j}$ for $j \in \{\phi, \psi\}$. This allows for subject-specific variation to

be incorporated in the temporal dynamics of the node and community embeddings. Finally, for the latent community assignment of each edge we define

$$q_{\lambda_z}(z_i^{(s,t)}|\phi_{w_i}^{(s,t)}, \phi_{c_i}^{(s,t)}) = \text{Categorical}(\pi_i^{(s,t)}) \quad (17)$$

$$\text{logit } \pi_i^{(s,t)} = \text{MLP}_{\lambda_z}(\phi_{w_i}^{(s,t)} \odot \phi_{c_i}^{(s,t)}) \quad (18)$$

where $\text{MLP}_{\lambda_z} : \mathbb{R}^{H_\phi} \to \mathbb{R}^K$ is $L_z$-layered MLP. In contrast to the generative model, the variational distribution of the community assignment now includes information from neighboring nodes via $c_i^{(s,t)}$. See Figure 2 for a graphical representation of the inference model.

**Training objective** Substituting the variational distribution from (11) and the joint distribution from (9) into the ELBO (10) gives the full training objective, which for the $s$-th subject is defined as

$$\mathcal{L}_{\text{ELBO}}^{(s)}(\theta, \lambda) =$$

$$\sum_{t=1}^{T} \sum_{i=1}^{E^{(s,t)}} \left( \mathbb{E}_{q_{\lambda_z} q_{\lambda_\psi}} \left[ \log p_{\theta_c}(c_i^{(s,t)}|w_i^{(s,t)}, \psi_{z_i}^{(s,t)}) \right] \right.$$

$$- \mathbb{E}_{q_{\lambda_\phi}} \left[ D_{\text{KL}}[q_{\lambda_z}(z_i^{(s,t)}|\phi_{w_i}^{(s,t)}, \phi_{c_i}^{(s,t)})|| \right.$$

$$\left. \left. p_{\theta_z}(z_i^{(s,t)}|\phi_{w_i}^{(s,t)})] \right] \right)$$

$$- D_{\text{KL}}[q_{\lambda_\alpha}(\alpha^{(s)})||p_{\theta_\alpha}(\alpha^{(s)})] \sum_{t=1}^{T} \left( \quad (19) \right.$$

$$- \sum_{n=1}^{V} \mathbb{E}_{q_{\lambda_\phi}} \left[ D_{\text{KL}}[q_{\lambda_\phi}(\phi_n^{(s,t)}|\phi_n^{(s,t-1)})|| \right.$$

$$\left. p_{\theta_\phi}(\phi_n^{(s,t)}|\phi_n^{(s,t-1)})] \right]$$

$$- \sum_{k=1}^{K} \mathbb{E}_{q_{\lambda_\psi}} \left[ D_{\text{KL}}[q_{\lambda_\psi}(\psi_k^{(s,t)}|\psi_k^{(s,t-1)})|| \right.$$

$$\left. \left. p_{\theta_\psi}(\psi_k^{(s,t)}|\psi_k^{(s,t-1)})] \right] \right)$$

where $D_{\text{KL}}[\cdot||\cdot]$ denotes the Kullback-Leibler (KL) divergence. By maximizing the ELBO, the parameters $(\theta, \lambda)$ of the generative model and inference model can be jointly learnt. See Figure 2 for a graphical representation of the inference model of NeuroEvolve.

---

**Algorithm 2** NeuroEvolve inference model

1: **Input:** Common node set $\mathcal{V}$, source nodes of all edges $\{w_i^{(s,t)} : i = 1, \dots, E^{(s,t)}\}_{s,t=1}^{S,T}$
2: **Require:** Number of communities $K \in \mathbb{N}$; embedding dimensions $H_\alpha$, $H_\phi$, $H_\psi \in \mathbb{N}$; number of layers in NNs $L_\phi, L_\psi$ $L_z \in \mathbb{N}$
3: $\mathcal{L} \leftarrow 0$
4: $\{\mu^{(s)}, \log \sigma^{(s)}\}_{s=1}^{S} \leftarrow \text{Normal}(\mathbf{0}_{H_\alpha}, \mathbf{I}_{H_\alpha})$
5: **repeat**
6:   **for** $s \leftarrow \text{RandomShuffle}[1 : S]$ **do**
7:     $q_{\lambda_\alpha}(\alpha^{(s)}) = \text{Normal}(\mu^{(s)}, \sigma^{(s)}\mathbf{I}_{H_\alpha})$
8:     $\alpha^{(s)} \sim q_{\lambda_\alpha}(\alpha^{(s)})$
9:     **for** $t \leftarrow 1$ to $T$ **do**
10:      **for** $k \leftarrow 1$ to $K$ **do**
11:       **if** $t = 1$ **then**
12:        $\psi_k^{(s,0)} = \text{MLP}_{\lambda_\psi}(\alpha^{(s)})$
13:       **end if**
14:       $\{\tilde{\mu}_k^{(s,t)}, \log \tilde{\sigma}_k^{(s,t)}\} = \text{GRU}_{\lambda_\psi}(\psi_k^{(s,t-1)})$
15:       $q_{\lambda_\psi}(\psi_k^{(s,t)}|\psi_k^{(s,t-1)}) =$
        $\text{Normal}(\tilde{\mu}_k^{(s,t)}, \tilde{\sigma}_k^{(s\,t)}\mathbf{I}_{H_\psi})$
16:       $\psi_k^{(s,t)} \sim q_{\lambda_\psi}(\psi_k^{(s,t)}|\psi_k^{(s,t-1)})$
17:      **end for**
18:      **for** $n \leftarrow 1$ to $V$ **do**
19:       **if** $t = 1$ **then**
20:        $\phi_n^{(s,0)} = \text{MLP}_{\lambda_\phi}(\alpha^{(s)})$
21:       **end if**
22:       $\{\hat{\mu}_n^{(s,t)}, \log \hat{\sigma}_n^{(s,t)}\} = \text{GRU}_{\lambda_\phi}(\phi_n^{(s,t-1)})$
23:       $q_{\lambda_\phi}(\phi_n^{(s,t)}|\phi_n^{(s,t-1)}) =$
        $\text{Normal}(\hat{\mu}_n^{(s,t)}, \hat{\sigma}_n^{(s,t)}\mathbf{I}_{H_\phi})$
24:       $\phi_n^{(s,t)} \sim q_{\lambda_\phi}(\phi_n^{(s,t)}|\phi_n^{(s,t-1)})$
25:      **end for**
26:      **for** $i \leftarrow 1$ to $E^{(s,t)}$ **do**
27:       $\text{logit } \hat{\pi}_i^{(s,t)} = \text{MLP}_{\lambda_z}(\phi_{w_i}^{(s,t)} \odot \phi_{c_i}^{(s,t)})$
28:       $q_{\lambda_z}(z_i^{(s,t)}|\phi_{w_i}^{(s,t)}, \phi_{c_i}^{(s,t)}) = \text{Categorical}(\hat{\pi}_i^{(s,t)})$
29:       $z_i^{(s,t)} \sim q_{\lambda_z}(z_i^{(s,t)}|\phi_{w_i}^{(s,t)}, \phi_{c_i}^{(s,t)})$
30:       $\text{logit } \tilde{\pi}_i^{(s,t)} = \text{MLP}_{\theta_c}(\psi_{z_i}^{(s,t)})$
31:       $p_{\theta_c}(c_i^{(s,t)}|z_i^{(s,t)}) = \text{Categorical}(\tilde{\pi}_i^{(s,t)})$
32:       $c_i^{(s,t)} \sim p_{\theta_c}(c_i^{(s,t)}|z_i^{(s,t)})$
33:      **end for**
34:     **end for**
35:     Compute gradients of $\mathcal{L}_{\text{ELBO}}^{(s)}(\theta, \lambda)$ w.r.t. $(\theta, \lambda)$
36:     Perform gradient-based updates for $(\theta, \lambda)$
37:     $\mathcal{L} \leftarrow \mathcal{L} + \frac{1}{S}\mathcal{L}_{\text{ELBO}}^{(s)}(\theta, \lambda)$
38:   **end for**
39: **until** $\mathcal{L}$ converges

---

**Inference and learning** In order to use efficient stochastic gradient-based optimization techniques (Robbins & Monro, 1951) for learning the parameters, the gradient of the ELBO (19) must be estimated with respect to $(\theta, \lambda)$. The main challenge is obtaining gradients of the variables under expectation, i.e. $\mathbb{E}_{q_*}[\cdot]$, since they are stochastic. To allow gradients to pass through these sampling steps, we use the reparameterization trick (Kingma & Welling, 2013; Rezende et al., 2014) for the normal distributions and the Gumbel-softmax trick (Jang et al., 2016; Maddison et al., 2016)

*Table 1.* Results for graph reconstruction (top) and dynamic link prediction (bottom). First and second-best results shown in **bold** and underlined. Results with a statistically significant difference from NeuroEvolve are marked *.

| Model | HCP | | UKB | |
|---|---|---|---|---|
| | NLL ($\downarrow$) | MSE ($\downarrow$) | NLL ($\downarrow$) | MSE ($\downarrow$) |
| VGAE | $5.857 \pm 0.017$ * | $0.051 \pm 0.002$ * | $5.851 \pm 0.027$ * | $0.061 \pm 0.002$ * |
| OSBM | $5.808 \pm 0.026$ * | $0.051 \pm 0.003$ * | $5.726 \pm 0.039$ * | $0.052 \pm 0.003$ * |
| VGRAPH | $\underline{5.569 \pm 0.046}$ * | $0.022 \pm 0.004$ * | $5.716 \pm 0.037$ * | $0.020 \pm 0.003$ * |
| VGRNN | $5.674 \pm 0.034$ * | $\underline{0.011 \pm 0.003}$ * | $\underline{5.649 \pm 0.035}$ * | $\underline{0.014 \pm 0.002}$ * |
| ELSM | $5.924 \pm 0.040$ * | $0.081 \pm 0.002$ * | $5.809 \pm 0.024$ * | $0.115 \pm 0.003$ * |
| NeuroEvolve | **4.587 ± 0.045** | **0.001 ± 0.002** | **4.586 ± 0.084** | **0.004 ± 0.003** |
| | AUROC ($\uparrow$) | AP ($\uparrow$) | AUROC ($\uparrow$) | AP ($\uparrow$) |
| VGAE | $0.661 \pm 0.010$ * | $0.674 \pm 0.008$ * | $0.688 \pm 0.010$ * | $0.607 \pm 0.009$ * |
| OSBM | $0.655 \pm 0.027$ * | $0.675 \pm 0.024$ * | $0.678 \pm 0.032$ * | $0.682 \pm 0.033$ * |
| VGRAPH | $\underline{0.689 \pm 0.004}$ * | $0.682 \pm 0.002$ * | $0.664 \pm 0.002$ * | $0.621 \pm 0.001$ * |
| VGRNN | $\underline{0.689 \pm 0.007}$ * | $\underline{0.698 \pm 0.006}$ * | $\underline{0.698 \pm 0.009}$ * | $\underline{0.696 \pm 0.007}$ * |
| ELSM | $0.669 \pm 0.004$ * | $0.662 \pm 0.002$ * | $0.661 \pm 0.001$ * | $0.662 \pm 0.002$ * |
| NeuroEvolve | **0.768 ± 0.026** | **0.732 ± 0.032** | **0.786 ± 0.040** | **0.762 ± 0.038** |

for the categorical distributions. This allows for the gradient of (19) w.r.t. $\theta$ and $\lambda$ to be easily computed via backpropagation (Rumelhart et al., 1986) making NeuroEvolve end-to-end trainable. In addition, we analytically calculate the KL terms for both normal and categorical distributions, which leads to lower variance gradient estimates and faster training as compared to noisy Monte Carlo estimates.

**Parameter sharing** We use the same NNs from the generative model to parameterize the variational distributions for the node and community embeddings as well as the edge community assignments. This not only spares additional trainable parameters for the variational distribution but also further links the variational parameters of $q_\lambda(\cdot)$ to generative parameters of $p_\theta(\cdot)$ resulting in more robust learning (Farnoosh & Ostadabbas, 2021). The set of parameters for the inference network is therefore $\lambda = \{\lambda_\alpha = \{\boldsymbol{\mu}^{(s)}, \boldsymbol{\sigma}^{(s)}\}_{s=1}^S, \lambda_\phi = \theta_\phi, \lambda_\psi = \theta_\psi, \lambda_z = \theta_z\}$.

## 5. Experiments

We evaluate NeuroEvolve against a range of unsupervised probabilistic graph representation learning baseline models on the tasks of graph reconstruction, dynamic link prediction, and graph classification. A good graph representation learning method should be able to preserve most of the original graph structure in it's embeddings. Since NeuroEvolve learns a hierarchy of embeddings, we use each task to quantitatively asses it's ability at preserving different levels of graph structure compared to baselines.

**Datasets** We construct two multi-subject DBG datasets using publicly available fMRI data from the Human Connectome Project (HCP) (Van Essen et al., 2013) and UK Biobank (UKB) (Sudlow et al., 2015). Both data sources represent well-characterized population cohorts that have undergone standardized neuroimaging and clinical assessments to ensure high quality. We randomly sample $S = 300$ subjects from each dataset ensuring an even split in biological sex. To create DBGs, we parcellate each image into $V = 360$ BOLD signals using the Glasser atlas (Glasser et al., 2016), apply sliding-window Pearson correlation (Calhoun et al., 2014) with a non-overlapping window of size and stride 30, and threshold the top $5\%$ values of each correlation matrix as connected following Kim et al. (2021). This results in $T = 16$ graph snapshots for each subject. Biological sex is taken as graph-level labels. We refer to Appendix C for further details on each dataset.

**Baselines** We compare NeuroEvolve against a range of different unsupervised probabilistic graph representation learning baseline models. For static baselines, we include variational graph autoencoder (VGAE) (Kipf & Welling, 2016b), a deep generative version of the overlapping stochastic block model (OSBM) (Mehta et al., 2019), and vGraph (VGRAPH) (Sun et al., 2019). For dynamic baselines we include variational graph recurrent neural network (VGRNN) (Hajiramezanali et al., 2019) and evolving latent space model (ELSM) (Gupta et al., 2019). Finally, for graph classification we include a support vector machine which takes as inputs static FC matrices (FCM) (Abraham et al., 2017). Further details about baseline models can be found in Appendix D.

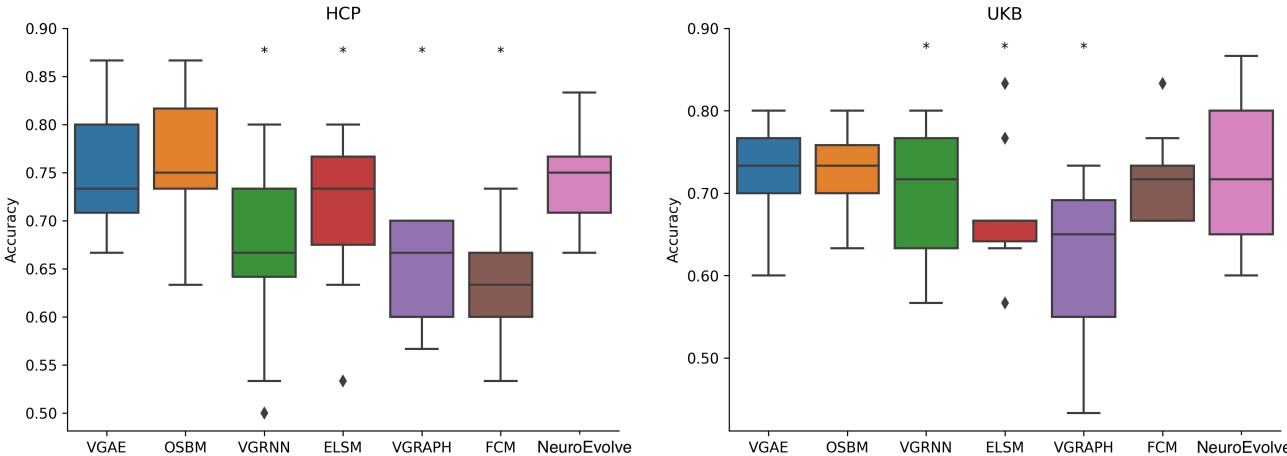

*Figure 3.* Graph classification results (5 runs). Results with a statistically significant difference from NeuroEvolve are marked *.

**Implementation** We split both datasets into 80/10/10% training/validation/test data along the snapshot dimension. All models are trained for 1,000 epochs using the Adam optimizer (Kingma & Ba, 2014) with decoupled weight decay (Loshchilov & Hutter, 2017). For static graph baselines VGAE, OSBM, VGRAPH we train on each snapshot independently and use node/community embeddings at the last training snapshot to make predictions. All models are trained 5 times using different random seeds and the model with the lowest validation negative log-likelihood (NLL) is saved. See Appendix E for further implementation details.

**Evaluation metrics** For graph reconstruction, we calculate the probability of observing edges over test snapshots using NLL and also compare the mean-squared error (MSE) between actual and reconstructed node degree over all snapshots. For dynamic link prediction, we sample an equal number of positive and negative edges and measure performance using area under the receiver operator curve (AUROC) and average precision (AP). Finally, for graph classification, we predict the biological sex for each subject's DBG and evaluate on accuracy. To do this, we average per-subject node-level embeddings for the baseline models and use the graph-level embeddings for NeuroEvolve before training a support vector machine (Murphy, 2012) using 10-fold cross-validation. Finally, for comparing results we use the almost stochastic order (ASO) test (Del Barrio et al., 2018; Dror et al., 2019) with significance level 0.05 and correct for multiple comparisons (Bonferroni, 1936).

## 6. Results

**Graph reconstruction and dynamic link prediction** Table 1 summarizes the test results averaged over 5 runs. From the results, it is clear that on both tasks NeuroEvolve outper-

forms all baselines by statistically significant margins. In particular, for graph reconstruction NeuroEvolve achieves an 18% and 30% relative improvement in NLL on HCP and UKB compared to the second-best baselines, respectively. For dynamic link prediction, the relative improvement of NeuroEvolve is $> 11\%$ in AUCROC and $> 5\%$ in AP compared to second-best baselines depending on dataset.

**Graph classification** For graph classification, NeuroEvolve achieves $\sim 75\%$ accuracy for HCP and $\sim 73\%$ for UKB (see Fig. 3). We outperform 4 baselines and show indiscernible performance to VGAE and OSBM. To show the interpretative power of NeuroEvolve, we re-run the graph classification experiment for HCP with the embeddings of each community separately. We find a community which comprises brain regions in the Cingulo-opercular (CON) and the Somatomotor (SMN) networks, which achieves 68% accuracy. This finding is in agreement with studies that show SMN is predictive of gender (Zhang et al., 2018). With the exception of VGRAPH, which NeuroEvolve outperforms, such an interpretability analysis cannot be done in a computationally feasible way by any of the other baselines.

## 7. Interpretability analysis

Evidence from fMRI studies suggests complex community structures exist within DBGs (Ting et al., 2020; Martinet et al., 2020b). These communities often correspond to groups of anatomically neighboring and/or functionally related brain regions that are engaged in specialized information processing.

**Community overlap** In order to interpret the community embeddings learnt by NeuroEvolve, for the $k$-th community embedding we create a node score vector $\bar{\psi}_k \in [0, 1]^V$ by

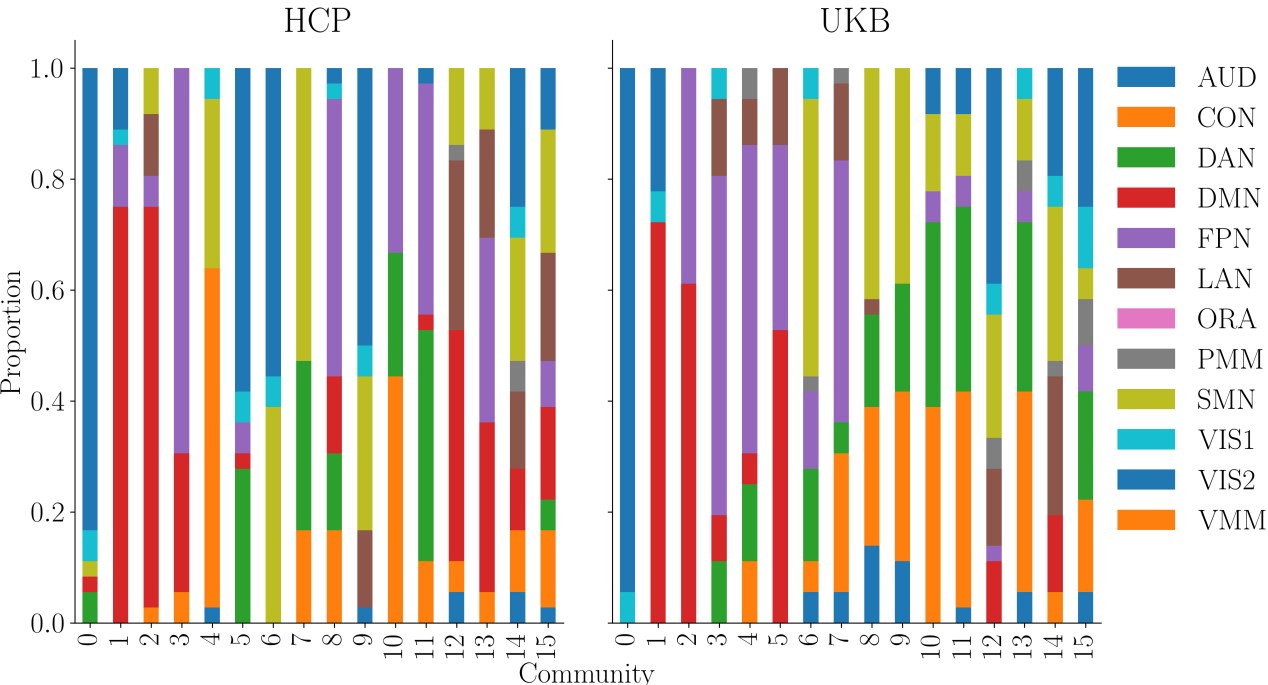

*Figure 4.* Overlap between node communities learned by NeuroEvolve and FCNs from Ji et al. (2019).

averaging sampled community embeddings over subjects and snapshots following

$$\bar{\psi}_k = \frac{1}{ST} \sum_{s=1}^{S} \sum_{t=1}^{T} \text{Softmax}\big(\text{MLP}_{\lambda_c}(\psi_k^{(s,\,t)})\big). \quad (20)$$

We keep the top $10\%$ highest scoring nodes in each score vector and compare their node composition, in terms of proportion of overlap, with known communities from neuroscience literature.

Figure 4 shows the proportion of overlap between nodes in each community and the nodes from FCNs described in Ji et al. (2019) (see Appendix B). It is clear that NeuroEvolve finds communities consisting of nodes that significantly overlap with existing FCNs. In particular, across HCP and UKB nodes in community labelled "0" almost fully correspond to the visual network (VIS1 and VIS2). This is in keeping with the image acquisition protocol of both datasets: subjects were required to keep their eyes open and fixed on a cross-hair. Remarkably, communities "1" and "2", the second and third most homogeneous communities across both datasets, corresponds largely to the default mode network (DMN), which is well known to dominate resting-state activity (Yeshurun et al., 2021). The inspection of additional communities, along with their evolution over time, has the potential to unveil the relationship between dynamic brain connectivity changes and brain disorders (Heitmann & Breakspear, 2017).

## 8. Conclusion

We propose NeuroEvolve, a hierarchical deep generative model designed for unsupervised representation learning of DBGs derived from fMRI data. Specifically, NeuroEvolve jointly learns distributions over graph-, community-, and node-level embeddings that evolve over time. Using these embeddings, NeuroEvolve is able to significantly outperform state-of-the-art baselines on the tasks of graph reconstruction and dynamic link prediction. Moreover, an analysis of the learnt dynamic community-node distributions shows significant overlap with existing FCNs from neuroscience literature further validating our method.

## Acknowledgments

We would like to thank the anonymous reviewers for their insightful comments and valuable feedback, which greatly contributed to improving the quality of this work. This research was supported by The Alan Turing Institute under the EPSRC grant EP/N510129/1. The data was provided [in part] by the Human Connectome Project, WU-Minn Consortium (Principal Investigators: David Van Essen and Kamil Ugurbil; 1U54MH091657) funded by the 16 NIH Institutes and Centers that support the NIH Blueprint for Neuroscience Research; and by the McDonnell Center for Systems Neuroscience at Washington University.

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

# A. Notation

Table 2. Summary of notation.

| Notation | Description |
|---|---|
| $S \in \mathbb{N}$ | Number of subjects. |
| $T \in \mathbb{N}$ | Number of snapshots. |
| $\mathcal{G}^{(1:S, 1:T)} = \{\mathcal{G}^{(s,t)}\}_{s,\,t=1}^{S,\,T}$ | Multi-subject dynamic brain graph (DBG) dataset derived from functional magnetic resonance imaging (fMRI). |
| $\mathcal{G}^{(s,t)} = (\mathcal{V}, \mathcal{E}^{(s,t)})$ | DBG of the $s$-th subject at the $t$-th snapshot. |
| $\mathcal{V} = \{v_1, \ldots, v_V\}$ | Set of common nodes. |
| $V \in \mathbb{N}$ | Number of nodes. |
| $\mathcal{E}^{(s,t)} \subseteq \mathcal{V} \times \mathcal{V}$ | Edge set. |
| $(w_i^{(s,t)}, c_i^{(s,t)}) \in \mathcal{E}^{(s,t)}$ | Source node and target node of the $i$-th edge. |
| $E^{(s,t)} \in \mathbb{N}$ | Number of edges. |
| $K \in \mathbb{N}$ | Number of communities. |
| $\boldsymbol{\alpha}^{(s)} \in \mathbb{R}^{H_\alpha}$ | Subject embedding of dimensionality $H_\alpha$. |
| $\boldsymbol{\phi}_n^{(s,t)} \in \mathbb{R}^{H_\phi}$ | Node embedding of dimensionality $H_\phi$. |
| $\boldsymbol{\psi}_k^{(s,t)} \in \mathbb{R}^{H_\psi}$ | Community embedding of dimensionality $H_\psi$. |
| $z_i^{(s,t)} \in [1:K]$ | Community assignment for the $i$-th edge. |
| $\Omega^{(s,t)} = \{\boldsymbol{\alpha}^{(s)}, \boldsymbol{\phi}^{(s,t)}, \boldsymbol{\psi}^{(s,t)}, \{z_i^{(s,t)}\}_{i=1}^{E^{(s,t)}}\}$ | Set of latent variables. |
| $p_\theta(\mathcal{G}^{(1:S, 1:T)}, \Omega^{(1:S, 1:T)})$ | Joint distribution of observed DBG and unobserved latent variables, i.e. generative model with parameters $\theta$. |
| $q_\lambda(\Omega^{(1:S, 1:T)} \mid \mathcal{G}^{(1:S, 1:T)})$ | Approximate posterior distribution, i.e. inference model with parameters $\lambda$. |
| $\sigma_j \in \mathbb{R}_{\geq 0}$ | Temporal smoothness hyperparameter for $j \in \{\phi, \psi\}$. |
| $\mathrm{MLP}_{\theta_*}(\cdot)$ | Multilayered perception (MLP) with $L_{\theta_*}$ layers and parameters $\theta_*$. |
| $\mathrm{GRU}_{\theta_*}(\cdot)$ | Gated recurrent unit (GRU) with $L_{\theta_*}$ layers and parameters $\theta_*$ |

# B. Functional connectivity networks

| Abbreviation | Functional connectivity network |
|---|---|
| AUD | Auditory network |
| CON | Cingulo-opercular network |
| DAN | Dorsal-attention network |
| DMN | Default mode network |
| FPN | Frontoparietal network |
| LAN | Language network |
| ORA | Orbito-affective network |
| PMM | Posterior-multimodal network |
| SMN | Somatomotor network |
| VIS1 | Visual network 1 |
| VIS2 | Visual network 2 |
| VMM | Ventral-multimodal network |

Table 3. Functional connectivity networks (FCNs) from Ji et al. (2019).

## C. Datasets

To create multi-subject DBG datasets, we use real fMRI scans of the brain from the UK Biobank (Sudlow et al., 2015) and Human Connectome Project (Van Essen et al., 2013). Both data sources represent well-characterized population cohorts that have undergone standardized neuroimaging and clinical assessments to ensure high quality.

**UK Biobank**[2] **(UKB)**   The UKB dataset consists of $S = 300$ resting-rate fMRI scans (i.e. 3D image of the brain taken over consecutive timepoints) randomly sampled from the v1.3 January 2017 release ensuring an equal male/female split (i.e. sex balanced) with an age range of $44 - 57$ years. The total number of images for each scan is $490$ timepoints (6 minutes duration with a repetition time of 0.74s). The dataset is minimally preprocessed following the pipeline described in Alfaro-Almagro et al. (2018).

**Human Connectome Project**[3] **(HCP)**   The HCP dataset similarly consists of $S = 300$ sex balanced resting-state fMRI scans randomly sampled from the S1200 release with an age range of $22 - 35$ years. Only images from the first scanning-session using left-right phase encoding are used. The total number of images for each scan is $1, 200$ timepoints (15 minutes duration with a repetition time of 0.72s). The dataset is minimally preprocessed following the pipeline described in Glasser et al. (2013)

**Further preprocessing**   The fMRI scans from each dataset are further preprocessed to create DBGs. First, each scan is transformed into a multivariate timeseries of BOLD signals using the Glasser atlas (Glasser et al., 2016) to average voxels within $V = 360$ brain regions. Next, to ensure comparability with UKB, we truncate the length of HCP timeseries to $490$ timepoints. Following the commonly used sliding-window method (Calhoun et al., 2014), we use Pearson correlation to calculate FC matrices within non-overlapping windows of length $1 < W \leq 490$ along the temporal dimension. At every window, we create an edge set of a unweighted and undirected graph with no self-edges by thresholding the top $1 \leq \epsilon < 100$ percentile values of the lower triangle of the FC matrix (excluding the principal diagonal) as connected following Kim et al. (2021). For both datasets, we choose $W = 30$ and $\epsilon = 5$ resulting in $T = \lfloor 490/30 \rfloor = 16$ graph snapshots each with $E^{(s, t)} = \lfloor (360(360 - 1)/2)(5/100) \rfloor = 3, 231$ edges.

## D. Baselines

We compare NeuroEvolve against a range of static and dynamic unsupervised probabilistic graph representation learning models, all with publicly available code. We leave comparisons to popular deterministic baselines such as Dynamic-Triad (Zhou et al., 2018), DySAT (Sankar et al., 2020), and DynNode2Vec (Mahdavi et al., 2018) for future work. Since all of the baselines were originally designed to model large single-subject dynamic graphs, we had to further adapt each implementation to work with multi-subject dynamic graphs.

**Variational graph auto encoder**[4] **(VGAE)** (Kipf & Welling, 2016b)   An extension of the variational autoencoder (Kingma & Welling, 2013) for graph structured data. Specifically, VGAE uses a graph convolutional network (GCN) (Kipf & Welling, 2016a) to learn a distribution over node embeddings. Originally designed for static graphs, we train VGAE on each dynamic graph snapshot independently.

**Overlapping stochastic block model**[5] **(OSBM)** (Mehta et al., 2019)   A deep generative version of the overlapping stochastic block model (Miller et al., 2009). In particular, OSBM places a stick-breaking prior over the number of communities which allows the model to automatically infer the optimal number of communities from the data during training. Similar to VGAE, OSBM uses a GCN to parameterize the distribution over node embeddings and is designed for static graphs.

**Variational graph recurrent neural network**[6] **(VGRNN)** (Hajiramezanali et al., 2019)   An extension of VGAE for dynamic graphs. Using a modified graph recurrent neural network, VGRNN is able to learn dependencies between and within

---

[2]https://www.ukbiobank.ac.uk
[3]https://www.humanconnectome.org
[4]https://github.com/tkipf/gae
[5]https://github.com/nikhil-dce/SBM-meet-GNN
[6]https://github.com/VGraphRNN/VGRNN

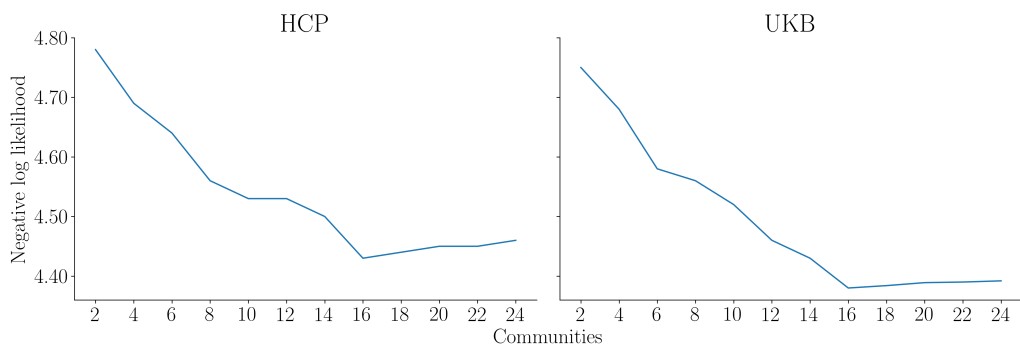

*Figure 5.* Elbow plot for finding the optimal number of communities $K$ based on validation NLL.

changing graph topology over time. Similar to NeuroEvolve, the prior distribution over node embeddings is parameterized using embeddings from previous snapshots.

**Evolving latent space model**[7] **(ELSM)** (Gupta et al., 2019)    A generative model for dynamic graphs that learns node embeddings and performs community detection. In particular, node embeddings are initially sampled from a Gaussian mixture model over communities and then evolved over snapshots using long-short term memory units (Hochreiter & Schmidhuber, 1997). Unlike the previous baselines, ELSM does not use a GNNs to parameterize model distributions.

**vGraph**[8] **(VGRAPH)** (Sun et al., 2019)    Similar to NeuroEvolve, VGRAPH simultaneously learns node embeddings and community assignments by modeling nodes as being generated from a mixture of communities. The generative process of VGRAPH also relies on edge information. Since VGRAPH only models static graphs, we train it on each dynamic graph snapshot independently.

## E. Implementation details

**Software and hardware**    All models were developed in Python 3.7 (Python Core Team, 2019) using scikit-learn 1.1.1 (Pedregosa et al., 2011), PyTorch (Paszke et al., 2019), and numpy 1.1.1 (Harris et al., 2020). Statistical significance tests were carried out using deep-significance 1.1.1 (Ulmer et al., 2022). Experiments were performed on a Linux server (Debian 5.10.113-1) with a NVIDIA RTX A6000 GPU with 48 GB memory and 16 CPUs.

**Hyperparameter optimization**    We use model and training hyperparameter values described in the original implementation of each baseline as a starting point for tuning on the validation dataset. Since searching for optional values for each hyperparameter configuration was outside the scope of the paper, we focused mainly on tuning the dimensions of NN hidden layers and embeddings, as well as the learning rate. For OSBM, VGRAPH and ELSM, we set the number of communities to the optimally tuned value of NeuroEvolve. To prevent overfitting, all models were trained using early-stopping with a patience of 15 based on the lowest validation NLL.

For NeuroEvolve, we choose the optimal number of communities $K = 16$ using the the elbo method (Thorndike, 1953) applied to validation NLL as shown in Figure 5. In the generative model, we fix the temporal smoothness hyperparameters $\sigma_\phi = \sigma_\psi = 0.01$. In the inference model, we fix the number of layers for all NNs to $L_\phi = L_\psi = L_z = 1$. For the Gumbel-softmax reparameterization trick we anneal the softmax temperature parameter starting from a maximum of 1 to a minimum of 0.05 at a rate of $3e$-4.

---

[7]https://github.com/sh-gupta/ELSM
[8]https://github.com/fanyun-sun/vGraph

