# OpenReview forum: "NeuroEvolve: A Dynamic Brain Graph Deep Generative Model"
_ICML.cc/2023/Workshop/IMLH — IMLH 2023 Poster_

### Official Review · Reviewer_TwaR · 2023-06-11
**Not very appropriate for this workshop**

**Rating:** 3
**Confidence:** 4

**Review:**

What stands out the most is the “interpretability” part of this paper. I understand the point of that section, to verify the communities discovered by DBGDGM are in line with well-established communities. However, I do not think this really furthers the field of XAI. This paper would be more appropriate for a different conference or workshop.
Figure 4 is also a little confusing. I cannot tell whether these communities actually match what is found in Ji. et. al. The authors should probably explain it a little bit better.
Finally, the accuracy in figure 3 doesn’t seem to show much improvement, if any, of the authors’ model over other generative models.

---

### Official Review · Reviewer_99Z7 · 2023-06-15
**Overall a good paper with few problems**

**Rating:** 9
**Confidence:** 5

**Review:**

Pros:
1. The idea is novel and the presentation is clear.
2. The proposed method is effective.
3. The writing is good.
Cons:
1. The reviewer would like to know why the proposed BDGDGM did not outperform VGAE and OSBM in Tab. 3 while significantly outperforming these baselines in Tab. 1.

---

### Official Review · Reviewer_H8tN · 2023-06-18
**The paper proposes a graph generative model for graph representation learning on dynamic graphs, which outperforms other static/dynamic baselines on downstream tasks like graph reconstruction, dynamic link prediction.**

**Rating:** 8
**Confidence:** 3

**Review:**

- quality: 4/5
- clarity: 3/5
- originality: 4/5
- significance: 4/5

# pros
The proposed DBGDGM model outperforms state-of-the-art baselines in various tasks and provides interpretability through the analysis of learnt community distributions.

# cons
- The definition of "community", and why we should focus on "community" for representation learning is unclear. Since this is the main difference from other dynamic baselines, such as VGRNN, it is necessary to clarify.
- The method part in the main paper is too long. The derivation of ELBO objective can be removed to the supplementary.
- The experiment result in Fig.4 is confusing. What does overlap refer to in this figure?

---

### Meta-Review · Program_Chairs · 2023-06-19

**Recommendation:** Accept (Poster)
**Confidence:** 4

**Metareview:**

This work received mixed review comments. Overall, this work is still interesting for the workshop. The authors are encouraged to incorporate the comments to improve the final version.

---

### Decision · Program_Chairs · 2023-06-20

Accept (Poster)